# A Systematic Review of Sleep–Wake Disorder Diagnostic Criteria Reliability Studies

**DOI:** 10.3390/biomedicines10071616

**Published:** 2022-07-06

**Authors:** Christophe Gauld, Régis Lopez, Pierre Philip, Jacques Taillard, Charles M. Morin, Pierre Alexis Geoffroy, Jean-Arthur Micoulaud-Franchi

**Affiliations:** 1Department of Child Psychiatry, Hospices Civils de Lyon, 69000 Lyon, France; christophe.gauld@chu-lyon.fr; 2UMR CNRS 8590 IHPST, Sorbonne University, 75007 Paris, France; 3Institut des Neurosciences de Montpellier (INM), University Montpellier, 34000 Montpellier, France; r-lopez@chu-montpellier.fr; 4Inserm, Unité des Troubles du Sommeil, Département de Neurologie, CHU Montpellier, 34000 Montpellier, France; 5University Sleep Clinic, Services of Functional Exploration of the Nervous System, University Hospital of Bordeaux, Place Amélie Raba-Léon, 33 076 Bordeaux, France; pr.philip@free.fr; 6CNRS, SANPSY, Université de Bordeaux, UMR6033, 33000 Bordeaux, France; jacques.taillard@u-bordeaux.fr; 7École de Psychologie, Université Laval, 2325 Rue des Bibliothèques, Québec City, QC G1V 0A6, Canada; cmorin@psy.ulaval.ca; 8Centre D’étude des Troubles du Sommeil, Université Laval, 2325 Rue des Bibliothèques, Québec City, QC G1V 0A6, Canada; 9Département de Psychiatrie et d’addictologie, AP-HP, GHU Paris Nord, DMU Neurosciences, Hopital Bichat—Claude Bernard, 75018 Paris, France; pierre.a.geoffroy@gmail.com; 10GHU Paris—Psychiatry & Neurosciences, 1 Rue Cabanis, 75014 Paris, France; 11NeuroDiderot, Inserm, Université de Paris, FHU I2-D2, 75019 Paris, France; 12CNRS UPR 3212, Institute for Cellular and Integrative Neurosciences, 67000 Strasbourg, France

**Keywords:** sleep–wake disorders, reliability, Cohen’s kappa coefficient, clinical significance, field trial, systematic review

## Abstract

The aim of this article is to provide a systematic review of reliability studies of the sleep–wake disorder diagnostic criteria of the international classifications used in sleep medicine. Electronic databases (ubMed (1946–2021) and Web of Science (—2021)) were searched up to December 2021 for studies computing the Cohen’s kappa coefficient of diagnostic criteria for the main sleep–wake disorder categories described in the principal classifications. Cohen’s kappa coefficients were extracted for each main sleep–wake disorder category, for each classification subtype, and for the different types of methods used to test the degree of agreement about a diagnosis. The database search identified 383 studies. Fifteen studies were analyzed in this systematic review. Insomnia disorder (10/15) and parasomnia disorder (7/15) diagnostic criteria were the most studied. The reliability of all sleep–wake disorders presented a Cohen’s kappa with substantial agreement (Cohen’s kappa mean = 0.66). The two main reliability methods identified were “test–retest reliability” (11/15), principally used for International Classification of Sleep Disorders (ICSD), and “joint interrater reliability” (4/15), principally used for Diagnostic and Statistical Manual of Mental Disorders (DSM) subtype diagnostic criteria, in particularl, the DSM-5. The implications in terms of the design of the methods used to test the degree of agreement about a diagnosis in sleep medicine are discussed.

## 1. Introduction

The diagnostic criteria of the main sleep–wake disorders have been defined by an iterative series of nosological proposals since the 1980s. The first nomenclature was used by Kales and Kales in 1974 in the New England Journal of Medicine [1] to organize the findings of the newly formed sleep research community. The first consensual classification was the Diagnostic Classification of Sleep and Arousal Disorders (DCSAD) proposed by the Association of Sleep Disorders Centers (ASDC) and the Association for the Psychophysiological Study of Sleep (APSS), and published in Sleep in 1979 [2]. Currently, there are three principal classifications of sleep–wake disorders: the International Classification of Sleep Disorders, Third Edition (ICSD-3) proposed by the American Academy of Sleep Medicine (AASM) [3], the sleep–wake disorders section of the Diagnostic and Statistical Manual of Mental Disorders, Fifth Edition (DSM-5) proposed by the American Psychiatric Association (APA) [4], and the International Classification of Diseases, Eleventh Revision (ICD-11) proposed by the World Health Organization [5]. One of the main aims of such classifications is to enhance the interrater reliability of diagnoses, which is the degree of agreement about a diagnosis among clinicians and/or over time [6,7,8]. More precisely, reliability could be considered as a signal-to-noise ratio indicator since there are three major sources of noise in clinical diagnostic practice: (i) the possible inconsistencies of expression of diagnostic criteria by patients during a clinical interview (e.g., the way the patient expresses his/her sleep complaint), (ii) the collection of information by the clinician (e.g., the way the clinician gives a labeling of the sleep complaint) [9], (iii) and the application of those criteria by the clinicians for a diagnosis at the end of a clinical interview process (e.g., the way the clinician makes diagnosis inference from the symptoms) [10]. These factors highlight the importance of both structured interviews and diagnostic algorithms. Cohen’s kappa coefficient is thought to be the most robust and most used of the reliability measures [11]. Different designs of methods can be used to test the degree of agreement about a diagnosis to compute Cohen’s kappa coefficient [6].

The study of the reliability of the diagnostic criteria of a classification is especially important since reliability constitutes a prerequisite for the main goals of sleep–wake disorder classifications in sleep medicine: (i) communication between patients, practitioners, and researchers, (ii) diagnostic, prognostic, and therapeutic decisions, (iii) diagnostic coding required by health care systems, and (iv) issues of validity and understanding of physiopathological mechanisms [12]. Despite the importance of the reliability of sleep–wake classification for multiple purposes relevant to research and clinical management, Edinger and Morin have noted, “currently, there is a paucity of data concerning the reliability of DSM and ICSD subtypes, and the data available are primarily to the insomnia spectrum disorders” [12].

Nevertheless, given that “the clinical interview is the most important component for any sleep disorder” [12], DSM and ICSD subtype diagnostic criteria have been associated with the development of standardized structured interviews that aim to systematically gather the relevant information for a diagnosis. These standardized structured interviews evaluate the presence of sleep–wake disorders based on questions asked of patients during a clinical interview, and most have been associated with evaluation of the reliability of sleep–wake disorder diagnoses. Two structured interviews have been particularly tested to evaluate principal sleep–wake disorders by the sleep research community. The Sleep-EVAL, developed by Ohayon, Guilleminault et al. in 1999 [13], constitutes an important step in the development of a structured and systematic clinical evaluation in sleep medicine [13]. The Sleep-EVAL was based on a mixture of DSM-IV and ICSD-1 (1990) diagnostic criteria. Following the Sleep-EVAL, the Diagnostic Interview for Sleep Patterns and Disorders (DISP) was developed by Merikangas et al. in 2014 [14] and was part of the PHENX project of the National Institutes of Health (NIH) to develop a consensus measure of phenotypes [15]. The DISP was based on the ICSD-2, with modifications performed by the authors [14].

In the psychiatric research community, it is interesting to note that each version of the DSM was associated with the construction of standardized structured interviews and evaluation of reliability following a research project associated with the revision of DSM diagnostic criteria called “field trials” [6,7,16,17]. In this way, the Structured Interview for Sleep Disorders (SIS-D), which was the earlier published structured interview for sleep medicine, was based on the DSM, Third Edition Revised (DSM-III-R) [18]. The Duke Structured Interview Schedule (Duke SIS), was based on the DSM, Fourth Edition Revised (DSM-IV-TR) and the second edition of the ICSD (ICSD-2) [19]. Finally, the Structured Clinical Interview for DSM-5 Sleep Disorders (SCISD) has accompanied development of the sleep–wake disorders section of the DSM, Fifth Edition (DSM-5) [20].

However, despite efforts to examine the reliability of these different structured interviews [12], no systematic review has been conducted to analyze the reliability of sleep–wake disorder diagnosis criteria of these principal classifications for diagnosing the main sleep–wake disorders used by sleep specialists (ICSD, DSM, and ICD). Yet, this is an important step for improving the relevance of sleep–wake disorder classifications in sleep medicine [12]. To address this issue, we conducted a systematic review aimed at centralizing reliability studies in the field of sleep medicine to summarize the different Cohen’s kappa coefficients for diagnostic criteria of the main sleep–wake disorder categories. This systematic review only focuses on clinical evaluations, and not on polysomnographic data. This analysis will also enable proposals for the design of methods used to test the degree of agreement about a diagnosis in sleep medicine.

## 2. Materials and Methods

PRISMA (Preferred Reporting Items for Systematic Reviews and Meta-Analyses) guidelines for systematic review and meta-analysis were followed [21].

### 2.1. Eligibility Criteria

The eligibility criteria of the reliability studies included in this article are the following.

Inclusion criteria were:Evaluation of sleep–wake disorder diagnosis criteria of the different versions of the DSM, ICSD, and ICD.Evaluation of diagnostic criteria based on clinical interviews or on clinical data recording.Reliability, defined as the degree of agreement among raters, measured by the Cohen’s kappa coefficient.Full original research papers, published in a peer-reviewed journal, suitable for inclusion in a systematic review and meta-analysis.Absence of restriction made on the date of articles included in this review.

Exclusion criteria were:Structured interview containing a single question on which reliability was assessed.Self-rated questionnaires developed for the patient to complete by him/herself to assess the presence of sleep symptoms.Absence of assessment of Cohen’s kappa coefficient.Studies not published in English.Studies investigating reliability only of polysomnography scoring.Single case studies, conference posters, reviews, meta-analyses, unpublished studies, opinions, and comments.

### 2.2. Search Strategy and Selection Process

The literature search was performed using PubMed and Web of Science databases up to December 2021. The search equation was built to include all sleep–wake disorders of the different versions of the ICSD, DSM, and ICD, and by adding the notion of reliability or agreement. We also added the notion of “field trials”, which were used for DSM reliability studies [7,8,17]. This equation is described in Appendix A (Appendix A).

After excluding duplicate references, two authors (CG and JAM) independently screened the title and abstract of each study identified by the search and applied the inclusion and exclusion criteria. Following this first step, the same procedure was followed for full texts of eligible studies. In case of disagreement between two evaluations, the article was discussed with a third author (RL). The reference lists of identified studies were reviewed for additional references. Furthermore, manual searches were conducted of all journals containing more than three of the studies identified via the database searches. These steps allowed us to reduce the possibility of missing non-indexed studies.

### 2.3. Data Extraction and Collection

From the included studies evaluating reliability of sleep–wake disorder diagnostic criteria, the following data were extracted (Appendix A):

General description

First Author/Year/Countries of evaluation/Journal;Monocentric/Multicentric studies;Type and number of different sleep–wake disorders evaluated;Classification(s) used for diagnostic criteria and year of publication (ICSD, DSM, ICD).

Modality of diagnostic evaluation

5.Number of clinical interviews performed with a patient, i.e., number of times a patient has been interviewed;6.Characteristics and number of raters, i.e., type of clinical profession and level of expertise in sleep medicine;7.Types of clinical interviews (structured, semi-structured, and unstructured) based on face-to-face or telephone interviews;8.Modality of diagnostic conclusion, with the interviewer clinical judgment (i.e., by the interviewer conducting the clinical interview or by the clinician based on clinical data recording) or without the interviewer clinical judgment (i.e., automatically);9.Reliability study quality standard criteria, i.e., reliability guidelines (e.g., Guidelines for Reporting Reliability and Agreement Studies (GRRAS)) [22].

Cohen’s kappa coefficient evaluation

10.Modality of diagnosis comparisons between raters, number of patients interviewed, and number of clinical judgment conclusions given;11.Number of Cohen’s kappa coefficients calculated;12.Cohen’s kappa coefficient of each of the different sleep–wake disorders studied and number of patients with the diagnosis.

### 2.4. Synthesis of Results

Descriptive statistics were calculated as frequencies (%) for categorical variables; means and standard deviations were computed for continuous variables.

The most frequent main sleep–wake disorder categories and the most frequent classification subtypes used were identified in the reliability studies included. The labelling of the six main sleep–wake disorder categories of the ICSD-3 was used (e.g., “psychophysiological insomnia” of the insomnia label of the ICSD-2 was merged under the ICSD-3 main category label of “insomnia disorder”; “confusional arousal” and “sleepwalking” were merged under the ICSD-3 main category label of “parasomnia disorder”). The six main sleep–wake disorder category labels were: insomnia disorders, central disorders of hypersomnolence, sleep-related breathing disorders, sleep-related movement disorders, circadian rhythm sleep–wake disorders, and parasomnia disorders. We identified the different designs of methods used to test the degree of agreement about a diagnosis to compute the Cohen’s kappa coefficient [6]. In particular, we considered the number of clinical interviews performed with a patient, the characteristics of the raters, the types of clinical interviews, and the modality of diagnostic comparison between raters. The aim was to identify a typology of methodologies of reliability studies used in sleep medicine, inspired by what has been reported in psychiatry in field trials of DSM reliability studies [6,7,17].

Concerning Cohen’s kappa coefficient evaluation, we drew a synthetic figure in which the different classification subtypes studied, ranging by year of publication, are presented on the abscissa, and each main sleep–wake disorder category on the ordinate. Each hexagon represents a study evaluating a given main sleep–wake disorder category. The size of each hexagon was computed as the mean of Cohen’s kappa coefficients for the sleep–wake disorders studied in one of the six main sleep–wake disorder categories (e.g., coefficients of “confusional arousal” of “sleepwalking” were averaged under the ICSD-3 label of “parasomnia disorder”). Different colors of each hexagon were used to represent the reliability methodology used, as previously identified. Lastly, we averaged Cohen’s kappa coefficients for the main sleep–wake disorder categories of each study. Landis and Koch provide guidelines for interpreting Cohen’s kappa coefficients, with values from 0.0 to 0.2 indicating slight agreement, 0.21 to 0.40 indicating fair agreement, 0.41 to 0.60 indicating moderate agreement, 0.61 to 0.80 indicating substantial agreement, and 0.81 to 1.0 indicating almost perfect or perfect agreement [23]. The figure was plotted with R and the online package plotly (https://plotly.com/ accessed on 21 December 2021). Finally, we looked for potential differences in the means of Cohen’s kappa coefficients between the main categories of sleep–wake disorders, each subtype of classification (ICSD, DSM, and ICD subtypes), the different methodologies to study reliability identified, and the modality of diagnostic conclusions with or without the interviewer clinical judgment.

Concerning reliability methods, we drew a synthetic figure in which the different reliability methods depend on sleep–wake disorders and principal classifications. This Sankey diagram was plotted with R and the package ggsankey. Moreover, we looked for potential differences in the percentage of methods used between the main categories of sleep–wake disorders and the subtype of classification (ICSD, DSM, and ICD).

Statistical analysis was performed with R Version 4.1.1 (R Foundation for Statistical Computing, Vienna, Austria). Categorical variables were analyzed by Chi-square test and continuous variables were analyzed by ANOVA. The significance level was set at *p* < 0.05 (two-tailed).

## 3. Results

### 3.1. Results of the Literature Search

As shown in Figure 1, the initial search returned 383 references after duplicates were removed. Following preliminary screening of the titles and/or abstracts, 67 were excluded. Among the references reviewed in detail, 15 independent studies were selected for systematic review [13,14,18,20,24,25,26,27,28,29,30,31,32,33,34]. The articles included in the review are detailed in Appendix A.

### 3.2. General Description

#### 3.2.1. Description of the Selected Studies

All the results are provided in Appendix A. The studies were published between 1993 and 2018. Most of the studies were conducted by American (9/15, 60%) and Italian (4/15, 26.7%) sleep centers and were rarely multicentric, with only two studies with more than two sleep centers: Schramm et al. [18] with three centers and Buysse et al. [24] with five centers, all from the same country (Germany and the USA, respectively). Five of the studies (33.3%) analyzed the reliability of multiple main sleep–wake disorder categories [13,14,18,20,24]. Ten studies (66.6%) assessed only one specific main sleep–wake disorder category [25,26,27,28,29,30,31,32,33,34]. The number of patients interviewed varied from 10 [34] to 225 [14] (mean = 104; median = 105) and the number of patients with a sleep–wake diagnosis for computation of Cohen’s kappa coefficient varied from 1 to 172 depending on the study (with 5 studies which did not report the exact number of patients).

#### 3.2.2. Main Sleep–Wake Disorders Categories and Principal Classifications

The most frequent main sleep–wake disorder category studied was insomnia disorder (10/15 (66.7%), including five studies that specifically assessed this disorder), followed by parasomnia disorders (7/15 (46.7%)), with one specifically studying the two different types of parasomnia disorder [32], one specifically studying REM sleep behavior disorder (RBD) [33], and one specifically studying disorder of arousal [30]. The third most studied sleep–wake disorder was sleep-related breathing disorders, in 5/15 studies (33.3%), which assessed obstructive sleep apnea syndrome (OSAS) [13,14,18,20,24]. However, none of these studies were specifically designed to evaluate the reliability of diagnostic criteria of OSAS. The other third most studied sleep–wake disorder category was central disorder of hypersomnolence in 5/15 studies (33.3%) [13,14,18,20,34]; one specifically studied narcolepsy [34]. Circadian rhythm sleep–wake disorders and sleep-related movement disorders were less studied (*n* = 3 and *n* = 4, respectively).

The 10 different classification subtypes used in the reliability studies were the ICSD (1, 1-R, 2, and 3), the DSM (III, III-R, IV, IV-R, and 5), and the ICD-10. We also included a study using the International Restless Legs Syndrome Study Group (IRLSSG) diagnostic criteria [35]. The most frequent classification studied was the ICSD-2 (4/15 (26.7%)), followed by the ICSD, the ICSD-R, and the DSM-IV-TR (each 3/15 (20%)). Four studies compared Cohen’s kappa coefficients based on different classification criteria subtypes [25,26,27,29].

#### 3.2.3. Modality of Diagnostic Comparisons among Raters

To compute the Cohen’s kappa coefficient, we identified four different methods that assessed the degree of agreement about a diagnosis. Note that all these methods compare diagnosis following a clinical interview, but differ in the way the rater makes the diagnosis and the type of clinical data evaluated.

The first group of methods (11/15 studies) involved interviewing a patient several times with a specific maximal time interval. These methods have been designed as “test–retest reliability” methods.
○In seven studies, the patient was interviewed by two interviewers (generally with a good level of expertise in sleep medicine), and reliability was calculated by comparing the sleep–wake disorders diagnosis reached independently by different interviewers (in green in Figure 2 and Appendix A) [18,24,26,28,30,31,32]. Types of clinical interviews varied (structured, semi-structured, or unstructured), but all were based on face-to-face interviews performed at two different times. In this methodology to study reliability, the number of clinical interviews performed with a patient corresponds to the number of diagnosis conclusions given by the interviewer, and the modality of diagnostic conclusion was done by the interviewer [18,24,26,30,31] and/or automatically based on the clinical data collected during the interviews [28,32].○In four studies, a patient was interviewed independently at two different times by a clinician and an interviewer with a structured questionnaire (in blue in Figure 2 and Appendix A). The clinician (sleep specialist) conducted an unstructured face-to-face interview [13,14] or semi-structured telephone interview [25,29]. The structured questionnaire was a standardized assessment based on DISP (*n* = 1) [14] or the Brief Insomnia Questionnaire (BIQ) (*n* = 2) [25,29] for telephone interviews and based on Sleep-EVAL (*n* = 1) for face-to-face interviews [13], conducted by a non-sleep specialist and non-clinician [14]. Reliability was calculated by comparing the sleep–wake disorder diagnosis reached by the clinician conducting the interview and the diagnosis reached automatically based on the questionnaire responses (in blue in Appendix A). In this methodology to study reliability, the number of interviews performed with a patient corresponds to the number of diagnosis conclusions given by the clinician conducting the interview and given automatically.In the second group of methods (4/15 studies), diagnostic criteria by different clinicians were applied based on previous clinical data recordings. “Clinical data recordings” refer to the collection of data from video (*n* = 2), audio (*n* = 1), and medical records (*n* = 1). These methods have been designed as “joint interrater reliability” methods.
○Two studies evaluated a single recording of a clinical interview between a patient and a sleep specialist (raters for Vignatelli et al., 2003 [33], and 17 raters for Vignatelli et al., 2022 [34]), which was subsequently evaluated by several clinicians to estimate reliability (in yellow in Figure 2 and Appendix A) [33,34]. All recordings were videotapes of semi-structured face-to-face interviews. Reliability was calculated by comparing the sleep–wake disorder diagnosis done by a different clinician watching the same videotapes. In this methodology to study reliability, the number of clinical interviews performed with a patient was inferior to the number of diagnosis conclusions given by the clinician based on videotapes. The methodology of Vignatelli et al., 2002 [34] followed the GRRAS [22].○Two studies were based on a patient being interviewed by an interviewer, which was subsequently evaluated by a clinician based on the recordings of the previous clinical interview [20,27] (in purple in Figure 2 and Appendix A). The recordings constituted the clinical data of the medical records following an unstructured face-to-face clinical interview conducted by a sleep specialist [27] or the audio recording of a structured interview following a clinical telephone interview conducted by a non-sleep specialist and non-clinician [20]. Reliability was calculated by comparing the sleep–wake disorder diagnosis reached by the clinician conducting the interview [27] or automatically [20], and by the clinician based on the recording clinical data [20,27]. In this methodology to study reliability, the number of clinical interviews performed with a patient was two times inferior to the number of diagnosis conclusions given.

### 3.3. Cohen’s Kappa Coefficients of the Main Sleep–Wake Disorder Categories

All but two studies [33,34] included several Cohen’s kappa coefficients. Thus, a total of 69 Cohen’s kappa coefficient values in all studies for all sleep–wake disorders studied are reported in Appendix A.

After relabeling and merging each sleep–wake disorder into the six main sleep–wake disorder categories of ICSD-3 labels, we retrieved 41 mean Cohen’s kappa coefficients in the 15 studies evaluated (as presented in Appendix A and Figure 2). The ICSD diagnostic criteria were studied in 19/41 retrieved Cohen’s kappa coefficients, and DSM diagnostic criteria were studied in 19/41 retrieved Cohen’s kappa coefficients. Two retrieved Cohen’s kappa coefficients were based on the ICD [25,29] and one was based on IRLSSG [28] diagnostic criteria.

For the six main sleep–wake disorder categories, the means of Cohen’s kappa coefficients were as follows:-Cohen’s kappa of insomnia disorders [14,18,20,24,25,26,27,29,31]: mean = 0.61 (substantial agreement) [23] (*n* = 17/41 Cohen’s kappa coefficients retrieved, with four studies evaluating the Cohen’s kappa of different classifications; standard deviation [SD] = 0.21).-Cohen’s kappa coefficient of central disorders of hypersomnolence, which was narcolepsy in the majority of cases [14,18,20,34]: mean = 0.64 (substantial agreement) [23] (*n* = 4/41 Cohen’s kappa coefficients retrieved; SD = 0.22).-Cohen’s kappa coefficient of sleep-related breathing disorders, which was obstructive sleep apnea syndrome in the majority of cases [13,14,18,20,24]: mean = 0.65 (substantial agreement) [23] (*n* = 5/41 Cohen’s kappa coefficient retrieved; SD = 0.19).-Cohen’s kappa coefficient of sleep-related movement disorders, which was restless legs syndrome in the majority of cases [14,18,20,28,32]: mean = 0.76 (substantial agreement) [23] (*n* = 5/41 Cohen’s kappa coefficients retrieved; SD = 0.17).-Cohen’s kappa coefficient of circadian rhythm sleep–wake disorders [14,18,20]: mean = 0.61 (substantial agreement) [23] (*n* = 3/41 Cohen’s kappa coefficients retrieved; SD = 0.16).-Cohen’s kappa coefficient of parasomnia disorders [13,14,18,20,30,32,33]: mean = 0.61 (substantial agreement) [23] (*n* = 7/41 Cohen’s kappa coefficients retrieved; SD = 0.32).

For this collection of six sleep–wake disorder categories, Cohen’s global kappa coefficient was at 0.65 (substantial agreement).

We did not observe any significant differences in mean Cohen’s kappa coefficient between the six main sleep–wake categories (*p* = 0.461; F = 0.879), between the 10 subtypes of classifications (*p* = 0.087; F = 3.09), or between ICSD and DSM classifications (*p* = 0.93; F = 0.007). We did not observe any significant differences in mean Cohen’s kappa coefficient between the four types of methods identified (*p* = 0.461; F = 0.879), and between the modality of diagnostic conclusions with or without the interviewer clinical judgment (*p* = 0.725; F = 0.087).

### 3.4. Reliability Methods for the Main Sleep–Wake Disorder Categories

We drew a comprehensive mapping of each reliability method, depending on sleep–wake disorders and classifications using a Sankey diagram (Figure 3).

We did not observe any significant differences in terms of percentage of types of methods used between the six main sleep–wake categories (*p* = 0.922; Chi-square = 8.06). However, we found a significant difference in terms of percentage of types of methods used between the 10 subtype classifications studied (*p* < 0.001; Chi-square = 68.30) and between ICSD or DSM classifications (*p* < 0.001; Chi-square = 12.88). Test–retest reliability methods (Method 1 (green) and Method 2 (blue)) were used more frequently for ICSD diagnostic criteria, and joint interrater reliability methods (Method 3 (yellow) and Method 4 (purple)) were more frequently used for DSM diagnostic criteria (Table 1).

## 4. Discussion

This first systematic review analyzed the reliability of sleep–wake disorder diagnosis criteria of the principal classification subtypes (ICSDs, DSMs, and ICDs). The reliability of the main sleep–wake disorder categories analyzed presents a Cohen’s kappa coefficient with substantial agreement [23]. The key findings of this study are as follows: (i) the different main sleep–wake disorder categories were not studied equally in terms of the reliability of these diagnostic criteria; (ii) the methodologies of studies evaluating reliability of diagnosis of sleep–wake disorders are heterogeneous and depend on the classification subtypes used. Given that “for many sleep disorders, research and practice remain greatly hampered by a lack of universally accepted and precise diagnostic criteria” [36], and given the importance of good reliability of sleep–wake disorder classification for clinical and research purposes [12], the present findings should be discussed carefully, and proposal for the design of reliability methods used to test the degree of agreement about a diagnosis in sleep medicine should be explicitly proposed.

### 4.1. Study of Reliability of Sleep–Wake Disorder Diagnosis Criteria

Although the six main sleep–wake disorder categories are represented, there are significant disparities in the study of reliability between these different categories. As found by Edinger and Morin [12], insomnia disorder is the sleep–wake disorder for which reliability of these diagnostic criteria was the most studied in the field of sleep medicine. One hypothesis regarding this interest could be that, unlike other disorders, insomnia disorders are based on the subjective complaint of patients, as assessed by a clinical interview, and are thus committed to systematically verifying its reliability. Insomnia disorder was studied in ten studies, with five studies specifically investigating this disorder [25,26,27,29,31]. Diagnosis criteria of each version of the ICSD subtypes (except ICSD-3) [3], each version of the DSM subtypes (including DSM-5) [4], and the ICD-10 have been investigated [37]. Such interest in the reliability of diagnostic criteria of insomnia disorders can be related to the fact that this sleep–wake disorder benefited from a great deal of interest from the AASM, which commissioned a project to develop a standard definition for the currently recognized insomnia disorder [36,38]. Interestingly, the four studies that investigated both the Cohen’s kappa coefficient for ICSD and DSM subtype diagnosis criteria specifically evaluated insomnia disorder [25,26,27,29]. Such comparative reliability approaches highlight the interest in comparing the diagnostic criteria reliability of the two main sleep–wake disorders’ classification (ICSD and DSM) to evaluate the concordance and discordance between them more accurately, given the fact that a difference in diagnostic criteria can lead to a major difference in diagnosis [39,40] and treatment decisions [41]. Moreover, the only study found in our systematic review that evaluated both the reliability and validity of diagnostic criteria was specifically for insomnia disorder [26], following the guidelines of convergent and discriminant validation [42]. Validity evaluation could be considered a complementary approach of reliability, as it consists of evaluating whether the diagnostic criteria lead to a pathophysiological based category [43]. As Edinger and Morin stated, “The validity of past and present sleep disorder nosologies is also rather limited and unimpressive at this juncture” [12], and our systematic review confirms that validity is rarely studied in conjunction with reliability evaluation.

Unexpectedly, parasomnia disorders were the second most studied main sleep–wake disorder category (7/15 studies), with three studies specifically investigating the two main different types of parasomnia disorders [30,32,33]. Diagnostic criteria of each version of the ICSD and the DSM subtypes (except DSM-IV and IV-TR) have been investigated. RBD was evaluated in four studies (4/15 (26.7%)) [13,14,32,33] with one specific study [33]. RBD was more frequently evaluated than disorders of arousal (2/15 (13.3%)) [30,32]. Nevertheless, disorders of arousal (sleepwalking, confusional arousal, and sleep terrors) are the sole sleep–wake disorder diagnosis criteria of the ICSD-3 that have been investigated in a reliability study [30]. Note that all the reliability studies for disorders of arousal and one for RBD (that was specifically designed for this specific sleep disorder) were conducted by Italian sleep centers [30,32,33]. With regard to the challenge of developing a clear definition of symptoms for parasomnia disorders, to propose useful clinical practice guidelines of evaluation [44,45], reliability studies with inter-cultural evaluation may be of interest for parasomnia disorders. Additionally, reliability studies using video-polysomnographic (vPSG) criteria to investigate more effectively how slow wave sleep interruptions are associated with unusual behaviors [46] could increase the validity of the diagnosis [45]. Development and evaluation of reliable diagnostic criteria for other unusual sleep-related behavior, in particular sleep-related psychogenic dissociative disorders [47], should also be encouraged.

Other diagnosis criteria of the main sleep–wake disorder categories have been little studied and the impact of evolution of their diagnosis criteria on reliability have not been systematically evaluated. Among sleep-related breathing disorders, the diagnosis criteria of OSAS for ICSD-1, ICSD-2, DSM-IV, and DSM-5 have been particularly studied. sleep-related breathing disorders are primarily studied with Sleep-EVAL [13], DISP [14], and SCISD questionnaires [20]. The reliability of ICSD-3 diagnosis criteria has not been investigated. This observation is not surprising; for instance, diagnosis of apnea and hypopnea should be validated with sleep recording. No study specifically evaluated these sleep–wake disorders, despite the fact that they benefit, as does insomnia disorder, from the interest of the AASM project to develop a standard definition for currently recognized sleep-related breathing disorders in adults [48]. It seems that there was more interest given to the rules for scoring respiratory events in sleep [49], the impact of different rules [50], and the reliability of the scoring [51,52,53]. Nevertheless, reliability studies of clinical diagnostic criteria may be encouraged, given the importance of the evaluation of clinical characteristics in delineating the normal and the pathological [54], and in evaluating the severity [55,56,57,58] of OSAS, as presented in the Baveno classification of OSAS [59], which requires rigorous investigation of clinical diagnostic criteria (e.g., non-restorative sleep, insomnia, or sleepiness complaints). The latter are important complementary criteria to the apnea and hypopnea index (AHI) recorded and scored with polygraphy or polysomnography, to capture clinically meaningful OSAS phenotypes [55].

Among central disorders of hypersomnolence, narcolepsy diagnosis criteria [13,14,18,34] for ICSD-1, ICSD-R, ICSD-2, and DSM-III-R, have been particularly studied. However, only one study was specifically dedicated to this disorder [34], conducted by Italian sleep centers. The study of Silber et al. [60] was excluded from this review, despite its high Cohen’s kappa coefficient (0.97), because the diagnostic criteria were not based on an international classification. We also excluded the studies of Folkert and Lopez because they were based only on sleep polysomnographic features [61,62]. Regarding the intensive work done on the definition of the clinical diagnostic criteria and on the validity of the delineation of different types of central hypersomnolence disorders [63,64,65], concomitant reliability studies should be encouraged.

Sleep-related movement disorders, examined by studying restless leg syndrome (RLS) diagnosis criteria, were investigated in four studies [14,18,20,28] for ICSD-1, DSM-III, and DSM-5 with one specific study based on the IRLSSG criteria [28]. We included these studies, also described by Edinger and Morin [12], because there is a close relationship between the diagnosis criteria of RLS in the ICSD-1 and the IRLSSG criteria, despite differences as discussed previously [66,67]. Interestingly, the IRLSSG has commissioned a project to develop a standard definition for currently recognized RLS [35,67]. In this way, reliability studies should be encouraged. Moreover, reliability for periodic limb movement disorder, frequently associated with RLS, has not been investigated. Sleep-related bruxism has been investigated only once for diagnosis criteria of ICSD-R [32]. Note that, at this time, this sleep disorder was included in the main category “other parasomnia”, whereas it was included in the main category “sleep-related movement disorder” since the ICSD-2. Despite discussion of the delineation of sleep-related bruxism [54,68,69,70], we did not find reliability studies on these recent diagnostic criteria, which should be encouraged.

Lastly, circadian rhythm sleep–wake disorders were investigated, without an analysis of subtypes disorders, for DSM-III [18] and DSM-5 [20] diagnostic criteria. Only delayed sleep–wake phase disorder was specifically investigated for reliability in terms of ICSD-2 diagnostic criteria [14]. Considering the use of sleep logs in the diagnostic criteria of ICSD-3, the reliability analysis of sleep diary investigation may be necessary to investigate the clinical reliability of such disorders [71,72].

To summarize, the diagnosis criteria of ICSD and DSM main sleep–wake disorder categories have been investigated systematically (for each evolution of diagnosis criteria) for insomnia disorders but less systematically for other categories, highlighting the fact that the AASM, despite an encouraging task force to define and revise diagnostic criteria [35,36,48], did not systematically promote the investigation of reliability of diagnosis criteria in development of the ICSD, at the difference of the APA for the DSM that promote the so-called and largely discussed “field trials”, i.e., reliability projects conducted for the diagnostic criteria of each version of the DSM [7,8,17,24]. Systematic analysis of diagnostic criteria of the six main sleep–wake disorder categories was conducted only for ICSD-2 [14] with evaluation of the DISP questionnaire; ICSD diagnostic criteria were evaluated for only four main sleep–wake disorder categories with the Sleep-EVAL questionnaire (insomnia disorder, sleep-related breathing disorders, central disorder of hypersomnolence, parasomnia disorders) [13]. The reliability of ICSD-3 diagnostic criteria was not investigated, except for disorder of arousal [45]. For DSM diagnostic criteria, systematic analysis of the six main sleep–wake disorder categories was conducted for the DSM-III-R with evaluation of the SIS-D [18], and for the DSM-5 with evaluation of the SCISD [20]. For the DSM-IV, the Cohen’s kappa coefficients of only two main sleep–wake disorder categories were reported in the study by Buysse et al. [24], who used unstructured interviews to investigate insomnia disorder and sleep-related breathing disorder.

### 4.2. Methodology of Sleep–Wake Disorder Reliability Studies

As the method can influence the estimation of diagnostic reliability [6], our systematic review studied factors related to methodological issues. We identified two main different methods.

The first method was based on the fact that diagnostic criteria are applied by different clinicians who independently evaluate the same patient over a specific maximal time interval [13,14,18,24,25,26,28,29,30,31,32]. Two subtypes of methods were identified; seven studies compared the evaluation of two clinicians conducting the same face-to-face interviews [18,24,26,28,30,31,32], and four other studies [13,14,25,29] used a modified version of this approach by comparing the rating of a clinician conducting an interview with an interview conducted by a non-specialist following a structured questionnaire. In two cases such methodologies lead to evaluating what has been called a “test–retest reliability” [7,17]. This methodology was used for ICSD diagnostic criteria reliability studies, particularly in the studies of Ohayon et al. [13] and Merikangas et al. [14].

The second method was based on the fact that diagnostic criteria are applied by different raters observing the same clinical data recording [20,27,33,34]. Two subtypes of methods were identified. Two studies compared the rating of raters evaluating the recording [33,34], although two others [20,27] used a modified version of this approach by comparing the rating of the rater evaluating the recording with the rating of the clinician conducting the clinical evaluation. In these cases, such methodologies lead to evaluating what is termed “joint interrater reliability” [7,17]. This methodology was used for DSM diagnostic criteria reliability studies, particularly in the study of Taylor et al. [20] (for DSM-5 diagnostic criteria).

Differences in methods can be associated with different Cohen’s kappa coefficient estimations [7,17]. In the field of psychiatric “field trials”, and at the difference of the study of Taylor et al. [20] on the DSM-5 sleep–wake disorder diagnostic criteria, DSM-5 field trials for mental disorders used “test–retest reliability” methods [7,17], which found a generally lower Cohen’s kappa coefficient than joint interrater reliability. In contrast to the study of Buysse et al. [24], which used DSM-IV sleep–wake disorder diagnostic criteria, DSM-IV field trials for mental disorders used “joint interrater reliability” methods [73,74,75], which found generally higher Cohen’s kappa coefficients than test–retest reliability. DSM-III field trials used a mixture of these two methods [8]. Differences in these methodologies have led to the postulate that the acceptable reliability level for DSM-5 classification was a Cohen’s kappa coefficient between 0.2 and 0.4 [10] and between 0.4 and 0.6 for the DSM-IV.

In our systematic review, we did not find differences in Cohen’s kappa coefficient means between the types of methodologies, possibly due to other sources of variability such as the use of a structured interview [6] and representativeness of the sample in terms of sleep–wake disorder samples [17]. Due to the heterogeneity of methods in the reliability studies included, and with only one study [34] reporting Guidelines for Reporting Reliability and Agreement Studies (GRRAS), further investigation of Cohen’s kappa coefficient variations was difficult in the field of sleep medicine. Thus, we discuss two challenges to promoting further studies to improve the design of methods used to test the degree of agreement regarding diagnoses in sleep medicine.

The first challenge is to reinforce the importance of the methodology and tools used to assess reliability in the sleep research community, to promote better reliability studies. Such methodological issues could reduce the disagreement between raters due to methodological issues, in particular due to clinicians or interviewers who conduct clinical interviews and/or patients who provide different information to different clinicians. The role of the AASM, which commissioned the project to develop a standard definition for currently recognized sleep–wake disorders, in particular for insomnia disorder [36] and sleep-related breathing disorders [48], or other initiatives, in particular for RLS [35,67], central hypersomnolence disorders [63], parasomnia disorder [44,76], and sleep-related bruxism disorder [69], should lead to a common reliability study to evaluate diagnostic criteria of multiple sleep–wake disorders in a representative sample in multiple sleep centers. Indeed, we show in the present systematic review that development of a standard definition initiative did not systematically result in reliability studies, except for insomnia disorder [36] and RLS [28]. The lack of similar methodology between sleep–wake disorders makes comparison of Cohen’s kappa coefficients between sleep–wake disorders difficult to interpret. Moreover, the two studies evaluating multiple sleep–wake disorders using structured hetero-questionnaires, Sleep-EVAL [13] and DISP [14], are encouraging for the field of sleep medicine, although the methodology appears to have been constructed to evaluate the reliability of the instrument rather than the test–retest or interrater reliability among clinicians’ evaluations per se.

The second challenge is to evaluate more accurately the source of the diagnosis of unreliability in our sleep clinical community, to know how to modify the diagnostic criteria. Such modification should then be evaluated to test whether the modification made to the diagnostic criteria was successful in improving the reliability of nosological classification evolution. A first phase could evaluate the original ICSD draft criteria evolution; a second phase could re-evaluate the selected revised criteria. Indeed, despite diagnostic criteria explicitly found in nosological classification, clinicians can keep their own diagnostic definition or can differently apply or interpret a set of diagnostic criteria. Writing such diagnostic criteria can be challenging. As we showed recently, the ICSD criteria can lack homogeneity in the way diagnostic criteria define a sleep–wake disorder [77]. In particular, the evaluation of reliability can be strongly related to the determination of diagnostic thresholds, especially with regard to the clinical significance criterion (CSC) and to objective diagnostic criteria obtained by polysomnography scoring.

Firstly, the CSC can be defined as a subjective evaluation of the level of distress or impairment induced by the symptom criteria (i.e., disturbing symptoms or significant consequences). Thus, the CSC can be central in evaluation of reliability; it can eliminate both potential false positives and negatives that could have been addressed by modifying the level of distress or impairment required by the symptom criteria. However, in classifications of sleep–wake disorders, the CSC is not present in all disorders. Moreover, the sleep community has not extensively discussed this question, apart from the psychiatric community [78]. Operationalizations of the CSC during field trials were carried out for the different versions of the DSMs [7,8,17]. Integrating the CSC with each disorder may minimize false positive diagnoses in situations in which the symptom criteria do not necessarily indicate pathology [78], and especially by elevating the level of required distress or impairment.

The evaluation of the CSC raises the issue of its (i) inconsistency with respect to the expression of diagnostic criteria by patients during a clinical interview; (ii) its difficulty to be collected by the clinician; (iii) and its application by the clinicians for a diagnosis at the end of a clinical interview process. However, the collection of the CSC is an important element of diagnostic interviews and algorithms. In our study, the question arises of how the CSC was applied in studies, with the interviewer’s clinical judgment, or automatically. Despite the lack of statistical difference between them, attention should be given to the operationalization of the CSC in future reliability studies for the future diagnoses of sleep–wake disorders.

In the ICSD-3, it was decided not to include the CSC criterion in all disorders we recently analyzed [77], e.g., sleep-related breathing disorders did not exhibit the CSC, using the Baveno classification instead [59]. In studies of sleep–wake disorder reliability, however, it seems important to consider the role of the CSC, without necessarily requiring its systematic integration into all sleep–wake disorders of the classification. Such investigation of the CSC could be related to the reliability investigation of the “patient reported outcomes measurement information system” (PROMIS), as was done for field trials of DSM-5, which investigated the reliability of cross-cutting symptom assessments [79], and particularly a sleep PROMIS. This study has considered a question on sleep (“Problems with sleep that affect sleep quality over all?”) that can be considered close to a CSC. Note that this study was not included in this systematic review because it did not investigate diagnosis criteria; only one question was asked based on the PROMIS [79]. For instance, in the DSM-5 field trials, based on this question, test–retest reliability was measured at 0.72 (0.69–0.74) for adults, which suggests good reliability of the CSC.

Secondly, in addition to this consideration for the CSC, it also seems important to consider objective criteria, which raise questions of reliability. Field trials in sleep medicine should therefore also be capable of evaluating the impact of evaluation of such objective criteria. Reliability here needs to be assessed, not only with respect to scoring, as conducted in many studies that investigated it in terms of the AASM scoring manual [49], but also with respect to interpretation of the PSG with regard to clinical diagnostic criteria and the final diagnosis of a sleep–wake disorder. In particular, sleep bruxism is an interesting discussed condition in the integration of objective criteria and CSC [54,80,81]. Considering the objective criterion represented by the PSG appears to be essential, thereby marking a difference between “DSM psychiatric field trials” and potential “ICSD field trials”, the latter classification could be based on objective markers for some sleep–wake disorders. However, among the three studies that assessed the reliability of the DSM [18,20,24], none relied on such an objective criterion. Conversely, clinicians in the studies that used Sleep-EVAL and DISP revised their diagnosis according to PSG [13,14].

### 4.3. Limitations

Several limitations of this study should be acknowledged. The first limitation is having principally considered Cohen’s kappa means by main sleep–wake disorder categories, by merging similar disorders defined on diagnostic criteria belonging to different principal classifications. This point can be problematic because the diagnostic criteria of an old classification are not necessarily the criteria of the ICSD-3. However, data of all Cohen’s kappa extracted are available on Appendix A to allow additional analyses.

Secondly, we did not find parameters that could explain the variability of Cohen’s kappa coefficients among studies, potentially due to methodological heterogeneities not captured by our analysis. However, we have discussed a number of potentially influential methodological issues in terms of the design of reliability studies for sleep medicine. Indeed, as part of potential future field trials in sleep medicine, it could be necessary to provide information regarding the source of diagnostic unreliability (e.g., by debriefing the clinicians regarding the reason for the disagreement) [9,82].

Thirdly, some studies were excluded from this systematic review. However, we included studies with criteria very close to principal classifications. For example, we included the study on RLS diagnostic criteria from Hening et al. [28] because of the criteria’s proximity to ICSD and the use of international consensual criteria. Conversely, we did not include the study of Silber et al. [60] on narcolepsy due to the lack of international consensus on these diagnosis criteria. We therefore propose that future studies may include criteria recognized by principal classifications and the subject of a consensus; this could lead to an increase in reliability. Sleep-EVAL based on a “mixture” of DSM-IV and ICSD-1 diagnostic criteria [13] and sleep-pattern based on ICSD-2 diagnostic criteria “with minor modifications” [14] should discuss the modifications chosen in more detail, which could have an important influence on diagnosis rate [39] and therapeutic decision [41].

Fourthly, one main difficulty was due to the fact that some studies, particularly the most important ones (e.g., the initial article presenting Sleep-EVAL), were poorly referenced automatically with our keywords relating to “reliability”, despite the high popularity and number of publications using this tool [13]. Thus, it is possible some studies were missed. However, we manually searched the references and reviewed the grey literature. Thus, this limit is not so much a difficulty related to the extraction of keywords for this systematic review but, rather, an issue for the transparent dissemination of the term “reliability”. We maintain that those studies should include the term “reliability” in their keywords.

Fifthly, we did not select articles with other agreement measures (e.g., intraclass correlation coefficient among different interviewers) [83]. However, Cohen’s kappa coefficient, as used in this study, remains the primary and most discussed measure of reliability in the international literature.

Finally, we excluded (systematic) reviews on sleep reliability, despite interesting results (e.g., [36]), due to the lack of empirical evaluation of reliability. These studies were, however, considered in the Discussion.

## 5. Conclusions

In this systematic review of the reliability of diagnosis criteria of sleep–wake disorders, we identified 15 studies that evaluated reliability. Although the six main sleep–wake disorder categories are represented with a Cohen’s kappa coefficient possessing substantial agreement (0.66), some sleep–wake disorders are over-represented (e.g., insomnia and parasomnia disorders) whereas others are under-represented despite their important prevalence and public health challenges (e.g., OSAS, IRLSS) or the importance of their diagnosis in terms of study of their pathophysiology (e.g., narcolepsy type 1 or 2, idiopathic hypersomnia). Inspired by the field of psychiatry, these results should encourage the development of “field trials” for future iterations of the ICSD and, in particular, consideration of both the subjective criteria of clinical significance and objective criteria such as the results of the PSG. Moreover, by anticipating integration of the results of continuous field trials, it could be interesting to develop an ICSD-3 live version (ICSD-3.1, ICSD-3.2, etc.) able to dynamically adapt the ICSD to the evidence.

## Figures and Tables

**Figure 1 biomedicines-10-01616-f001:**
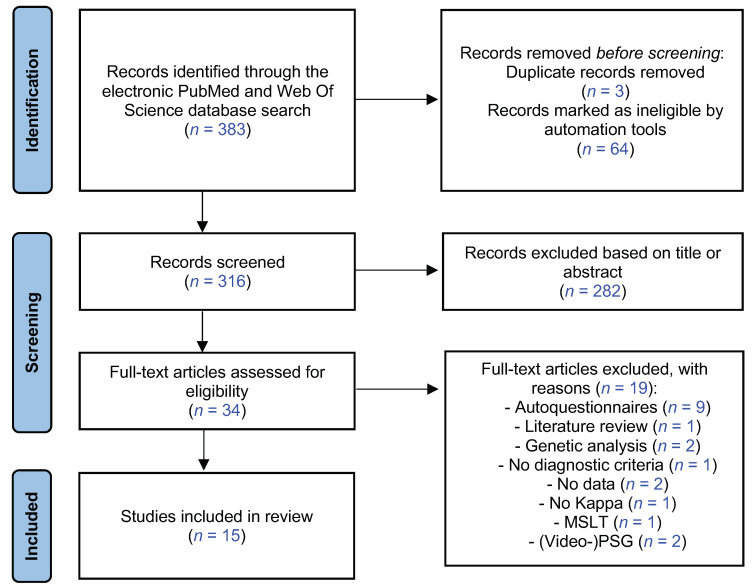
PRISMA flow diagram for the systematic review. MSLT, Multiple Sleep Latency Test [21].

**Figure 2 biomedicines-10-01616-f002:**
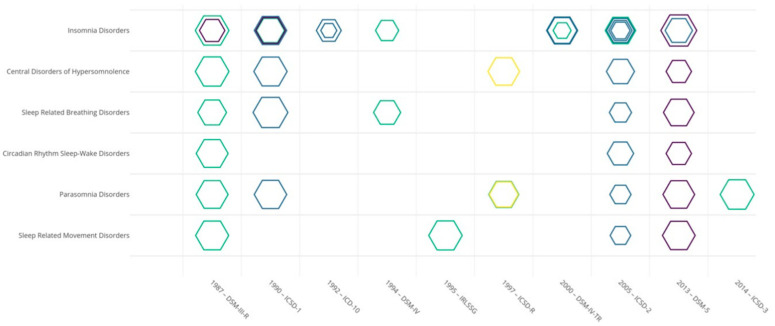
Graphical representation of 41 mean Cohen’s kappa coefficients retrieved (size of hexagons) organized in ordinate by the main sleep–wake disorder categories studied and in abscissa by the different classification diagnostic criteria studied, and ranging by year of publication. The size of the hexagons indicates the importance of the Cohen’s kappa coefficient: the larger the hexagon, the greater the Cohen’s kappa coefficient. Colors depend on the four reliability study methodologies identified. Method 1 (green) and Method 2 (blue) represent the “test–retest reliability” methods. Method 3 (yellow) and Method 4 (purple) represent the “joint interrater reliability” methods. The 69 Cohen’s kappa coefficients are not shown since the sleep–wake disorders were merged according to their current ICSD-3 main sleep–wake disorder category label. An interactive figure showing the references of the articles in which the Cohen’s kappa coefficient was retrieved is available at: https://chart-studio.plotly.com/~ChristopheGauld/48 (accessed on 21 December 2021).

**Figure 3 biomedicines-10-01616-f003:**
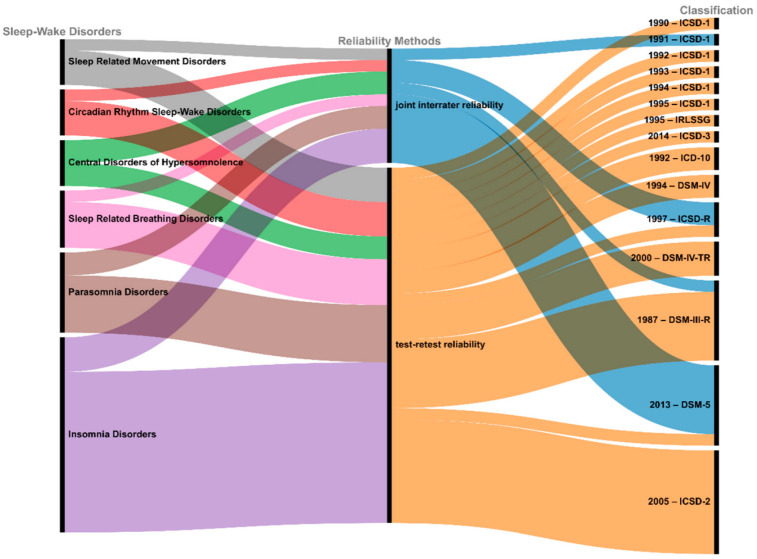
Sankey diagram of sleep–wake disorders, reliability methods, and classifications. The different reliability methods with their associated sleep–wake disorders and classifications are represented via flows. Nodes are represented as black rectangles, and the height represents their value. The width of each curved line is proportional to its values. Each column of a Sankey diagram needs to be read in pairs, i.e., sleep–wake disorders with reliability methods, and reliability methods with classifications. For instance, it is not possible to assert that ICSD-2 is only related to insomnia disorders via the method of test–retest reliability. To analyze the link between classification and sleep–wake disorders, we refer to Figure 2.

**Table 1 biomedicines-10-01616-t001:** Comparisons of reliability methods and subtypes of classifications. *n* represents the number of Cohen’s kappa coefficients identified for main sleep–wake disorders in the 15 studies included.

	Methods 1 and 2 “Test–Retest Reliability”	Methods 3 and 4 “Joint Interrater Reliability”	Total
DSMs	*n* = 3 (18% *) (kappa mean = 0.67)	*n* = 16 (75% *) (kappa mean = 0.70)	19
ICSDs	*n* = 14 (82% *) (kappa mean = 0.72)	*n* = 5 (25% *) (kappa mean = 0.67)	19
Total	17	21	38 **

* Percentage relative to the type of method. ** Only 38 Cohen’s kappa coefficients were retrieved, as previously described in Figure 2; Cohen’s kappa coefficients of the ICD (*n* = 2) and IRLSSG (*n* = 1) were not included.

## Data Availability

Not applicable.

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
