# Peer review of "A Systematic Review of Sleep–Wake Disorder Diagnostic Criteria Reliability Studies"

_biomedicines, 2022, doi:10.3390/biomedicines10071616_

Round 1

Reviewer 1 Report

Discuss the effects of Feature ranking methods on classification results

Authors should list the classification algorithms used in the previous studies, and how the results can be affected by the type of the classifier.

Discuss how the expert's decisions could  affect the reliability of proposed models.

Please improve the quality of figures.

Please refer to the all datasets used in the previous studies, and what are the channels used in classification.

Author Response

Discuss the effects of Feature ranking methods on classification results

Authors: We thank the reviewer for emphasizing the importance of methods on classification diagnostic results. The modalities of diagnostic conclusion are linked to two methods: with the interviewer clinical judgment (i.e., by the interviewer conducting the clinical interview or by the clinician based on clinical data recording) or without the interviewer clinical judgment (i.e., automatically). We now indicate this important information concerning the methods on classification results in the introduction, method, results and discussion.

Page 5, Lines 177-179: “Modality of diagnostic conclusion, with the interviewer clinical judgment (i.e., by the interviewer conducting the clinical interview or by the clinician based on clinical data recording) or without the interviewer clinical judgment (i.e., automatically).”

Page 6, Lines 224-228: “Finally, we looked for potential differences in the means of Cohen’s kappa coefficients between the main categories of Sleep-Wake disorders, each subtype of classification (ICSD, DSM, and ICD subtypes), the different methodologies to study reliability identified, and the modality of diagnostic conclusions with or without the interviewer clinical judgment.”

Authors should list the classification algorithms used in the previous studies, and how the results can be affected by the type of the classifier.

Authors: This information are indeed potentially very important. Classification algorithms were not found explicitly in the articles retrieved for this systematic review. This is an interesting prospect for further reliability studies. Thus, we sustain that the algorithms of the systems which give the diagnoses automatically should be freely accessible and explained in the reliability studies, which is not currently the case. We have now indicated these points in the revised version of the manuscript, in the introduction and the discussion.

Page 2, Lines 57-64: “More precisely, reliability could be considered as a signal-to-noise ratio indicator since there are three major sources of noise in clinical diagnostic practice: i) the possible inconsistencies of expression of diagnostic criteria by patients during a clinical interview (e.g., the way the patient expresses her/his complaint), ii) the collection of information by the clinician (e.g., the way the clinician gives a labeling) [9], iii) and the application of those criteria by the clinicians for a diagnosis at the end of a clinical interview process (e.g., the way the clinician makes diagnosis inference from the symptoms) [10]. These factors highlight the importance of both structured interviews and diagnostic algorithms.”

Pages 16-17, Lines 655-663: “The evaluation of the CSC raises the issue of its i) inconsistency with respect to the expression of diagnostic criteria by patients during a clinical interview; ii) its difficulty to be collected by the clinician; iii) and its application by the clinicians for a diagnosis at the end of a clinical interview process. However, the collection of the CSC is an important element of diagnostic interviews and algorithms. In our study, the question arises to know how the CSC was applied in studies with the interviewer clinical judgment, or automatically. Despite the lack of statistical difference between them, attention should be given to the operationalization of the CSC in future reliability studies for the future diagnoses of Sleep-Wake disorders.”

Discuss how the expert's decisions could affect the reliability of proposed models.

Authors: We thank the reviewer for this clarification because the importance of expert decisions can indeed have an influence on reliability. In the reliability studies included, some are based on a diagnosis given automatically, some are based on expert’s decision. In order to test how the expert’s decisions could affect the reliability, we carried out a comparison between the Kappa obtained from studies having given the diagnosis automatically or not. We have now indicated the analyze of Kappa between automatically and expert’s-based diagnosis in the revised manuscript.

Page 11, Lines 397-399: “We did not observe any significant differences in mean Cohen’s kappa coefficient between the four types of methods identified (p = 0.461; F = 0.879), and between the modality of diagnostic conclusions with or without the interviewer clinical judgment (p = 0.725; F = 0.087).

Please improve the quality of figures.

Authors: Figures are updated to improve quality. Note however that high quality figures are not inserted in the manuscript, but provided separately.

Please refer to the all datasets used in the previous studies, and what are the channels used in classification.

Authors: We thank the reviewer for this comment. As underlined by the reviewer, our limitation is to have merged similar diagnoses belonging to different classifications. This point can be problematic because the criteria of an old classification are not necessarily the criteria of the ICSD-3.

Thus, we highlight this point in a limitation added in the revised manuscript. Moreover, the details of these information are provided in Supplementary Materials.

Page 19, Lines 694-699: The first limitation is having principally considered Cohen’s kappa means by main Sleep-Wake disorder categories, by merging similar disorders defined on diagnostic criteria belonging to different principal classifications. This point can be problematic because the diagnostic criteria of an old classification are not necessarily the criteria of ICSD-3. However, data of all Cohen’s kappa extracted are available on Table 1 to allow additional analyzes.”.

Reviewer 2 Report

This is very interesting and useful systematic review for clinicians.

It was a pleasure to read this manuscript.

However, I found a few minor flaws:

1. The PRISMA 2020 Statement is available. Therefore Authors have to  report the systematic review in accordance to the updated Statement. Please remember to use updated flow diagram. Link to the PRISMA 2020 Statement

https://www.bmj.com/content/372/bmj.n71?gclid=EAIaIQobChMI-aP3-5Xn9wIVlNeyCh1ujAzGEAAYASAAEgIADvD_BwE

2. Authors wrote that Sleep Medicine is a new discipline (1974). However They provided an information in abstract and manuscript body that electronic databases PubMed and Web of Science were searched between 1946–2021. Does that make any sense? Please clarify this information.

3. In section 4.1 part about AHI Authors omit important paper: Malhotra A, Ayappa I, Ayas N, Collop N, Kirsch D, Mcardle N, Mehra R, Pack AI, Punjabi N, White DP, Gottlieb DJ. Metrics of sleep apnea

severity: beyond the apnea-hypopnea index. Sleep. 2021 Jul 9;44(7):zsab030. doi: 10.1093/sleep/zsab030. This article should be discuss in mentioned section.

4. In section 4.1 part about Sleep Related Bruxism Authors omit two important papers:

Lavigne G, Kato T, Herrero Babiloni A, Huynh N, Dal Fabbro C, Svensson

P, Aarab G, Ahlberg J, Baba K, Carra MC, Cunha TCA, Gonçalves DAG,

Manfredini D, Stuginski-Barbosa J, Wieckiewicz M, Lobbezoo F. Research

routes on improved sleep bruxism metrics: Toward a standardised

approach. J Sleep Res. 2021 Oct;30(5):e13320. doi: 10.1111/jsr.13320.

Wieczorek T, Wieckiewicz M, Smardz J, Wojakowska A, Michalek-Zrabkowska

M, Mazur G, Martynowicz H. Sleep structure in sleep bruxism: A

polysomnographic study including bruxism activity phenotypes across

sleep stages. J Sleep Res. 2020 Dec;29(6):e13028. doi:

10.1111/jsr.13028. Both articles should be discuss in mentioned section.

5. Authors have to increase a resolution of Figure 3. Currently the  quality of this Figure is very weak.

Author Response

This is very interesting and useful systematic review for clinicians.

It was a pleasure to read this manuscript.

Authors: We are very grateful to the reviewers for her/his encouragement.

However, I found a few minor flaws:

  1. The PRISMA 2020 Statement is available. Therefore Authors have to report the systematic review in accordance to the updated Statement. Please remember to use updated flow diagram. Link to the PRISMA 2020 Statement

https://www.bmj.com/content/372/bmj.n71?gclid=EAIaIQobChMI-aP3-5Xn9wIVlNeyCh1ujAzGEAAYASAAEgIADvD_BwE

Authors: We thank the reviewer for this vigilance. However, the PRISMA 2020 Statement has been up to date, regarding the flow diagram and the reference. We have used the most recent PRISMA Flow Diagram found on the official website:

http://prisma-statement.org/prismastatement/flowdiagram.aspx.

We have now indicated this point in the method of the revised manuscript.

Page 3, Lines 119-120 : “PRISMA (Preferred Reporting Items for Systematic reviews and Meta-Analyses) guidelines for systematic review and meta-analysis were followed [21].”

  1. Authors wrote that Sleep Medicine is a new discipline (1974). However They provided an information in abstract and manuscript body that electronic databases PubMed and Web of Science were searched between 1946–2021. Does that make any sense? Please clarify this information.

Authors: As rightly pointed out by the reviewer, this discrepancy may seem contradictory. We keep the research over a relatively long time to have a complete representativeness of the articles discussing reliability of sleep disorder diagnosis.

However, we do not wish to attempt to define the beginnings of Sleep Medicine. We have removed in the revised manuscript the notion that sleep medicine is a new discipline born in 1974.

The sentence “Sleep medicine is a relatively recent discipline in the history of medicine” has been removed.

  1. In section 4.1 part about AHI Authors omit important paper: Malhotra A, Ayappa I, Ayas N, Collop N, Kirsch D, Mcardle N, Mehra R, Pack AI, Punjabi N, White DP, Gottlieb DJ. Metrics of sleep apnea severity: beyond the apnea-hypopnea index. Sleep. 2021. This article should be discuss in mentioned section.

Authors: We thank the reviewer for this suggestion, which indeed represents an important reference. We had previously suggested this article in the OSAS discussion (Page 13, line 510), but we agree with the reviewer that a brief discussion about it may be in order. We therefore added a brief discussion about it.

Page 14, Lines 512-514: The latter are important complementary criteria to the Apnea and Hypopnea Index (AHI) recorded and scored with polygraphy or polysomnography, to capture clinically meaningful OSAS phenotypes [55].”

  1. In section 4.1 part about Sleep Related Bruxism Authors omit two important papers:

Lavigne G, Kato T, Herrero Babiloni A, Huynh N, Dal Fabbro C, Svensson P, Aarab G, Ahlberg J, Baba K, Carra MC, Cunha TCA, Gonçalves DAG, Manfredini D, Stuginski-Barbosa J, Wieckiewicz M, Lobbezoo F. Research routes on improved sleep bruxism metrics: Toward a standardized approach. J Sleep Res. 2021 Oct;30(5):e13320. doi: 10.1111/jsr.13320.

Wieczorek T, Wieckiewicz M, Smardz J, Wojakowska A, Michalek-Zrabkowska M, Mazur G, Martynowicz H. Sleep structure in sleep bruxism: A polysomnographic study including bruxism activity phenotypes across sleep stages. J Sleep Res. 2020 Dec;29(6):e13028. Both articles should be discuss in mentioned section.

Authors: We thank the reviewer for this absolutely important reference in an still open debate about the metrics, classification and categories of Sleep Bruxism. These two references are now added to the revised manuscript.

Page 18, Lines 684-685:In particular, Sleep Bruxism is an interesting discussed condition in the integration of objective criteria and CSC [54,80,81].

  1. Authors have to increase a resolution of Figure 3. Currently the quality of this Figure is very weak.

Authors: The quality of this figure has been revised. Note however that high quality figures are not inserted in the manuscript, but provided separately.

The English has been reviewed by a specialist.

Round 2

Reviewer 1 Report

authors have addressed my comments.